# Two Alimentary Canal Proteins, Fo-G_N_ and Fo-Cyp1, Act in Western Flower Thrips, *Frankliniella occidentalis* TSWV Infection

**DOI:** 10.3390/insects14020154

**Published:** 2023-02-03

**Authors:** Falguni Khan, David Stanley, Yonggyun Kim

**Affiliations:** 1Department of Plant Medicals, Andong National University, Andong 36729, Republic of Korea; 2Biological Control of Insects Research Laboratory, USDA/ARS, 1503 S Providence Road, Columbia, MO 65203, USA

**Keywords:** cyclophilin, FISH, *Frankliniella occidentalis*, glycoprotein, immunofluorescence assay, RNAi, TSWV

## Abstract

**Simple Summary:**

Tomato spotted wilt virus (TSWV) is a plant virus that causes significant economic loss to high-valued crops, including hot pepper and tomato. It is transmitted by some thrips species, including *Frankliniella occidentalis*. Thrips obtain TSWV by feeding on infected host plants. The TSWV infects the midgut and then enters the thrips bodies, where the virus translocates to salivary glands for subsequent transmission. Here we assessed the actions of two intestinal proteins, glycoprotein (Fo-G_N_) and cyclophilin (Fo-Cyp1) in their functional association with TSWV infection of the midgut. Silencing the two genes encoding these two proteins led to a near-zero reduction of TSWV in midguts and salivary glands of the infected thrips. We propose that the two intestinal proteins, Fo-G_N_ and Fo-Cyp1, are TSWV entry targets that are necessary to infect *F. occidentalis* for continued transmission to additional host plants.

**Abstract:**

Tomato spotted wilt virus (TSWV) is a plant virus that causes massive economic damage to high-valued crops. This virus is transmitted by specific thrips, including the western flower thrips, *Frankliniella occidentalis.* TSWV is acquired by the young larvae during feeding on infected host plants. TSWV infects the gut epithelium through hypothetical receptor(s) and multiplies within the cells for subsequent horizontal transmission to other plant hosts via the salivary glands during feeding. Two alimentary canal proteins, glycoprotein (Fo-G_N_) and cyclophilin (Fo-Cyp1), are thought to be associated with the TSWV entry into the gut epithelium of *F. occidentalis*. Fo-G_N_ possesses a chitin-binding domain, and its transcript was localized on the larval gut epithelium by fluorescence in situ hybridization (FISH) analysis. Phylogenetic analysis indicated that *F. occidentalis* encodes six cyclophilins, in which Fo-Cyp1 is closely related to a human cyclophilin A, an immune modulator. The Fo-Cyp1 transcript was also detected in the larval gut epithelium. Expression of these two genes was suppressed by feeding their cognate RNA interference (RNAi) to young larvae. The RNAi efficiencies were confirmed by the disappearance of the target gene transcripts from the gut epithelium by FISH analyses. The RNAi treatments directed to Fo-G_N_ or Fo-Cyp1 prevented the typical TSWV titer increase after the virus feeding, compared to control RNAi treatment. Our immunofluorescence assay using a specific antibody to TSWV documented the reduction of TSWV in the larval gut and adult salivary gland after the RNAi treatments. These results support our hypothesis that the candidate proteins Fo-G_N_ and Fo-Cyp1 act in TSWV entry and multiplication in *F. occidentalis*.

## 1. Introduction

Tomato spotted wilt virus (TSWV) is a plant virus, genus Orthotospovirus and family Bunyaviridae [1]. It is one of the most widely distributed plant viruses, transmitted by specific thrips mostly in genus *Frankliniella*, such as *F. occidentalis* and *F. schultzei* [2]. It is an enveloped spherical-shaped virus particle, and its genome consists of large (L), medium (M), and small (S) RNA segments [3]. The L segment encodes an RNA-dependent RNA polymerase in antisense. The M and S segments encode genes in ambisense. The M segment encodes two glycoproteins (G_N_ and G_C_) and a non-structural protein (NS_M_), and the S segment encodes a non-structural protein (NSs) in sense and a nucleocapsid protein (N) in antisense [4,5].

The Western flower thrips (WFT), *Frankliniella occidentalis*, is a thysanopteran. It transmits TSWV in a persistent, propagative way [6,7]. Because it has a wide host range and is distributed globally, it is a highly effective vector [8]. Young larvae acquire the virus, and the subsequent adults transmit it during feeding on host plants [9,10].

We are working to understand the process of TSWV transmission at the molecular level. In the viral entry phase, TSWV virions must move across the midgut apical membrane of the brush border, which may be mediated via an interaction with TSWV glycoproteins and a midgut receptor in a hypothetical endocytotic pathway [11,12]. Soluble viral G_N_ binds to the thrip’s midgut epithelium and prevents the interaction between TSWV and the thrip’s midgut [13]. Using this binding affinity of the viral G_N_, six *F. occidentalis* TSWV-interacting proteins (TIPs) were discovered by immunoblotting and subsequent proteomics [14]. Two TIPs, an endocuticle structural glycoprotein (Fo-G_N_) and a cyclophilin (Fo-Cyp1), are colocalized with the viral G_N_, indicating they may act in viral entry into a host insect. However, the physiological function of the candidate proteins in the viral entry is not yet clearly understood. In our approach to this issue, we tested the hypothesis that the candidate proteins Fo-G_N_ and Fo-Cyp1 act in TSWV entry and multiplication in *F. occidentalis*. Here, we report on the outcomes of experiments designed to test our hypothesis.

## 2. Materials and Methods

### 2.1. Insect Rearing

Western flower thrips, *F. occidentalis*, used in this study were obtained from a laboratory colony of the National Academy of Agricultural Sciences (Jeonju, Republic of Korea). The adults and larvae were reared on sprouted bean seed kernels (*Phaseolus coccineus*) that had germinated for 5 days prior to feeding [15]. Our standard conditions in the rearing room are 25 ± 2 °C, a 16L:8D photoperiod and 65 ± 5% relative humidity. A circular breeding container (100 mm × 40 mm, SPL, Pyeongtaek, Republic of Korea) was used to rear thrips from eggs to adults.

### 2.2. Bioinformatics

The *F. occidentalis* glycoprotein (Fo-G_N_; GenBank accession number: MH884757.1) and cyclophilin (Fo-Cyp1; GenBank accession number: MH884760.1) were analyzed. The obtained sequences were annotated with the BlastN program (http://www.ncbi.nlm.nih.gov, accessed on 29 November 2022). Phylogenetic analyses and phylogenetic tree construction were performed with the Neighbor-joining method using MEGA6.06 (www.megasoftware.net, accessed on 29 November 2022), and ClustalW programs, respectively. Bootstrapping values were obtained with 1000 repetitions to support each branch node in the phylogenetic tree. The protein domains were predicted using EMBL-EBI (www.ebi.ac.uk, accessed on 29 November 2022) and Pfam (http://pfam.xfam.org, accessed on 29 November 2022); the UCSF Chimera (http://www.cgl.ucsf.edu/chimera/, accessed on 29 November 2022) was used for searching protein motifs and superimposition analysis. Pymol (http://pymol.org, accessed on 29 November 2022) was used for active binding site analysis. A Rebers and Riddiford (RR) consensus was identified using CuticleDB (https://cuticledb.eesi.psu.edu/, accessed on 29 November 2022).

### 2.3. RNA Extraction, cDNA Synthesis, RT-PCR, and RT-qPCR

RNA samples were collected (~50 each, larvae, pupae, males, and females) for developmental expression analysis. For these samples, total RNAs were extracted using the Trizol reagent (Invitrogen, Carlsbad, CA, USA) according to the manufacturer’s instruction. After extraction, RNA was resuspended in nuclease-free deionized distilled water and quantitated using a spectrophotometer (NanoDrop, Thermo Scientific, Wilmington, DE, USA). The RNA was used for cDNA synthesis using an RT Premix (Intron Biotechnology, Seoul, Republic of Korea) containing an oligo-dT primer based on the manufacturer’s instructions. RT-PCR was performed using DNA Tag polymerase (GeneALL, Seoul, Republic of Korea) under these conditions: initial denaturation at 94 °C for 5 min, followed by 35 cycles at 94 °C for 1 min, selected annealing temperatures for 1 min, 72 °C for 1 min, and final extension at 72 °C for 10 min with gene-specific primers (Appendix A). Each RT-PCR reaction mixture (25 µL) consisted of a cDNA template, dNTP (each 2.5 mM), 10 pmol for each forward and reverse primer, and Taq polymerase (2.5 unit/µL). The stably expressed reference gene, elongation factor, *EF1*, was used as a reference gene. All gene expression levels in this study were determined using a Real-time PCR system (Step One Plus Real-Time PCR System, Applied Biosystem, Singapore) using Power SYBR Green PCR Master Mix (Life Technologies, Carlsbad, CA, USA) according to the guidelines of Bustin et al. [16]. RT-PCR was performed in a 20 µL reaction volume containing 10 µL of Power SYBR Green PCR Mix, 2 µL of cDNA template (60 ng/µL), and 1 µL each of forward and reverse primers (Appendix A). RT-PCR cycling began with a 95 °C heat treatment for 10 min and was followed by 35 cycles of denaturing at 95 °C for 1 min, selected annealing temperatures for 1 min, and extension at 72 °C for 1 min. The quality of the PCR products was assessed by melting curve analysis. Quantitative analysis was performed using the comparative CT (2^−∆∆CT^) method [17], where each experiment was replicated three times with biologically independent sample preparation.

### 2.4. dsRNA Preparation and RNA Interference (RNAi)

RNAi was performed using gene-specific dsRNA, prepared using a MEGAscript RNAi kit (Ambion, Austin, TX, USA) according to the manufacturer’s instructions. Briefly, gene-specific primers were prepared by adding the T7 sequence (TAATACGACTCACTATAGGGAGA) at the 5′ end. The PCR product was used as a template to generate dsRNA. Sense and antisense RNA strands were synthesized using T7 RNA polymerase at 37 °C for 4 h. A control dsRNA (dsCON) was also prepared by synthesizing a 520 bp fragment dsRNA of *CpBV302*, a viral gene. The resulting dsRNA was purified and mixed with transfection reagent Metafectene PRO (Biontex, Planegg, Germany) at a ratio of 1:1 (*v*/*v*), and then incubated at 25 °C for 30 min to form liposomes. The prepared dsRNA was applied to larvae using a feeding protocol and RNAi efficiencies were measured at 0, 6, 12, and 24 h by RT-qPCR as just described. Each treatment was replicated three times.

### 2.5. Immune Challenge with TSWV

TSWV was extracted from hot pepper (*Capsicum annum* L.) leaves, with visible ringspot symptoms, that were collected from a greenhouse in Andong, Korea. TSWV symptoms in leaves were confirmed with an Immunostrip TSWV kit (Agdia, Elkhart, IN, USA). About 5 g of plant tissues were ground in 1 mL filter-sterilized Phosphate buffer saline (PBS, pH 7.4) and centrifuged at 14,000× *g* for 5 min. The supernatant was used to make a viral suspension. For the *F. occidentalis* immune challenge, sprouted bean seed kernels were dipped into 1 mL of the viral suspension for 5 min and dried for 10 min under aseptic conditions. Treated bean seeds were placed in a circular breeding container (100 mm × 40 mm, SPL) and test insects were fed for 12 h, and then the virus-treated bean seeds were replaced with the sprouted beans for the insects to be used in experiments.

### 2.6. Immunofluorescence Assay

To identify possible TSWV midgut infections, L1 larvae were fed with gene-specific dsRNA. After 12 h of dsRNA feeding, when gene expression was suppressed, new beans soaked in TSWV-containing solution were applied as a diet. Twelve h later, the entire alimentary canals were isolated onto glass slides. The midgut preparations were fixed with 4% formaldehyde for 1 h at room temperature (RT). After washing with PBS 3X, the midgut was permeabilized with 1%Triton X-100 in PBS for 2 h at RT. After washing with PBS 3X, the preparations were blocked with 10% BSA in PBS at RT for 30 min. A TSWV antibody (Kerafast, Inc., Boston, MA, USA) was added to the gut and incubated at RT for 2.5 h. After washing with PBS 3X, midgut samples were incubated with 1% anti-rabbit-FITC conjugated antibody (Thermo Fisher Scientific, Seoul, Republic of Korea) in PBS at RT for 2 h. After washing with PBS 3X, the midguts were incubated with 4′,6-diamidino-2-phenylindole (DAPI, 1 μg/mL) (Thermo Fisher Scientific, Seoul, Republic of Korea) in PBS at RT for 5 min, for nucleus staining. After washing with PBS 3X and adding 50% glycerol, samples were covered by a cover glass (18 × 18 mm, Marienfeld, Königshofen, Germany) and observed with a fluorescence microscope (DM2500; Leica, Wetzlar, Germany) at 200× magnification.

### 2.7. Fluorescence In Situ Hybridization (FISH) Assay

The first instar larvae were treated with RNAi and TSWV as just described. Twelve h later, the guts were isolated onto a sterilized slide glass and fixed with 4% paraformaldehyde for 1 h at RT. After washing with 1× PBS, the midgut was permeabilized with 1%Triton X-100 in PBS for 2 h at RT. After washing with 1× PBS, the guts were rinsed in 2× sodium saline citrate (SSC) and incubated at 42 °C with 25 μL of pre-hybridization buffer (2 μL yeast tRNA, 2 μL 20× SSC, 4 μL dextran sulfate, 2.5 μL 10% SDS, and 14.5 μL deionized H_2_O) in dark and humid conditions for 1 h. Then pre-hybridization buffer was replaced with hybridization buffer (5 μL deionized formamide and 1 μL fluorescein-labeled oligonucleotide in 19 μL of the pre-hybridization buffer). DNA oligonucleotide probes were labeled at the 5′ end with fluorescein amidite (FAM) and purified on high performance liquid chromatography (Bioneer, Daejeon, Korea). For Fo-Cyp1, a probe (anti-sense) (5′-FAM-TTT-CTGGAAGTTCTCGTCAGCGA-3′) complementary to the mRNA and a negative control probe (sense) (5′-FAM-AAA-TCGCTGACGAGAACTTCCAG-3′) and for Fo-G_N_ an anti-sense probe (5′-FAM-TTT-GTCGAACTCCTGAACCTGGC-3′) and a sense probe (5′-FAM-AAA-GCCAGGTTCAGGAGTTCGAC-3′) were synthesized. The slides were covered with an RNAse-free cover slip and kept in a humid chamber at 42 °C overnight (for 16–17 h). After hybridization, the gut was washed twice with 4× SSC for 10 min each and incubated with 4× SSC containing 1%Triton X-100 in RT for 5 min. After washing with 4× SSC 3X, midgut samples were incubated at 37 °C with 1% anti-rabbit-FITC conjugated antibody (Thermo Fisher Scientific) in PBS under dark conditions for 30 min. After incubation, the midgut was washed twice with 4× SSC for 10 min each and once with 2× SSC and allowed to air dry. After adding a drop of 50% glycerol and incubating at RT for 15 min, samples were covered by cover glass and the slides were observed under a fluorescence microscope (DM2500) at ×200 magnification.

### 2.8. Indirect Enzyme-Linked Immunosorbent Assay (ELISA)

Indirect ELISA was conducted to detect TSWV in dsRNA-fed insect samples. The protein was extracted from insect samples at a 1/20 dilution with a coating buffer (0.05 M sodium carbonate, pH 9.6, containing 0.01% sodium azide). The mouse monoclonal TSWV antibody (Agdia) was diluted with conjugate buffer (PBST containing 2% PVP-40 and 0.2% BSA) and was used at a 1/8000 dilution. The alkaline phosphatase (AP)-conjugated goat anti-mouse IgG (Sigma-Aldrich Korea) was used at a 1/5000 dilution as the secondary antibody. AP substrate tablets (*p*-nitrophenyl phosphate disodium salt hexahydrate) (BCIP/NBT, Sigma-Aldrich Korea) dissolved in substrate buffer (9.7% diethanolamine and 0.02% NaN_3_, pH 9.8) were used for color development. The absorbance at 405 nm was determined after a 10 to 40 min incubation.

### 2.9. Statistical Analysis

All studies were performed using one-way ANOVA using the PROC GLM of SAS program [18]. Means were compared with the least squared difference (LSD) test. Each experiment was conducted with three replicates and plotted as mean ± standard error using Sigma plot.

## 3. Results

### 3.1. Molecular Characters and Expression Profile of Fo-G_N_ in F. Occidentalis

Figure 1a indicates that Fo-G_N_ is very similar to G_N_ counterparts in lepidopterans, hymenopterans, and coleopterans. Figure 1b shows the protein domain structure of Fo-G_N_, with a chitin-binding domain, which features a RR1 consensus.

Figure 2a indicates substantial Fo-G_N_ expression occurs in larvae, with lower expression in pupae. Within larvae, we recorded substantial expression in whole larvae and in isolated larval alimentary canals and significantly diminished expression in the remaining gut-free larval tissues (Figure 2b). FISH analysis using FAM-labeled probes shows Fo-G_N_ expression occurred in the midgut and salivary gland with the anti-sense probe, but not with the sense probe (Figure 2c).

### 3.2. Silencing Fo-G_N_ Led to Curtailed TSWV Multiplication

Larvae were treated with RNAi as described in the Section 2 and Fo-G_N_ expression was significantly reduced over the following 24 h (Figure 3a). Fo-G_N_ expression was not detectable in the midgut or salivary gland by FISH analysis (Figure 3b). RT-qPCR and ELISA analysis confirmed that TSWV multiplication in the midgut and salivary gland was reduced in the absence of the glycoprotein gene (Figure 3c). The immunofluorescence assay shows the larval stage TSWV signal did not appear in the alimentary canals of RNAi-treated larvae (Figure 3d). These treated larvae developed to adults and TSWV-associated staining was assessed (Figure 3e). In control adults, the TSWV signal was clear in the salivary gland and midgut. However, little was observed in the adults of the RNAi treatment.

### 3.3. Molecular Characters of Fo-Cyp1 and Its Expression Profile in F. occidentalis

Six cyclophilin genes (Fo-Cyp1~Fo-Cyp6) are annotated in the *F. occidentalis* (Figure 4). Phylogenetic analysis of the cyclophilin genes with selected *Drosophila* species and *Homo sapiens*, confirmed that our targeted cyclophilin gene is Fo-Cyp1 (Figure 4a). Domain analysis demonstrated the Fo-Cyp1 gene contains a cyclophilin-like domain similar to the human cyclophilin A and the *Drosophila melanogaster* cyclophilin 1 (Figure 4b). Three-dimensional modeling indicates the *F. occidentalis* and *Homo sapiens* proteins are very similar structurally (match score 704.9), as are the *F. occidentalis* and *Drosophila melanogaster* proteins (match score 772.9) (Figure 4c). By sequence alignment, the cyclosporin binding site occurs in *F. occidentalis*, based on *Homo sapiens* and *Drosophila melanogaster* (Figure 4d).

mRNA expression analysis of the indicated developmental stages show that Fo-Cyp1 is expressed at significantly higher levels in females compared to larvae, pupae and males (Figure 5a). Within the larval stage, mRNA expression was significantly higher in whole larvae than in the gut-removed remaining tissues. There were no significant expression differences between whole larvae and the isolated gut (Figure 5b). FISH analysis shows Fo-Cyp1 expression in the midgut and salivary gland. Fo-Cyp1 expression was detected in the midgut and salivary gland with the anti-sense probe, but not with the sense probe (Figure 5c).

### 3.4. Effect of Fo-Cyp1 RNAi on TSWV Multiplication

After feeding on a dsRNA suspension, Fo-Cyp1 gene expression was monitored at 0, 6, 12 and 24 h post treatment. At 12 h post feeding, Fo-Cyp1 expression was substantially reduced by approximately 2.5-fold (Figure 6a). After dsCyp1 application, cyclophilin expression was undetectable in midgut and salivary gland with the anti-sense, nor with the sense probes (Figure 6b). To determine whether the TSWV multiplication is related to Fo-Cyp1 in the midgut and salivary gland, dsRNA fed larvae were treated again with TSWV. RT-qPCR and ELISA data show that TSWV multiplication was reduced to near zero by 72 h post RNAi treatment *(*Figure 6c). Similarly, Figure 6d,e show the RNAi treatments led to very large reductions in immunofluorescence in larvae and adults.

## 4. Discussion

The data reported in this paper solidly support our hypothesis that the candidate proteins, Fo-G_N_ and Fo-Cyp1, act in TSWV entry and multiplication in *F. occidentalis*. Several points are germane. First, the *F. occidentalis* Fo-G_N_ gene is very similar to counterpart genes in 13 other species. Second, Fo-G_N_ is expressed in larvae, pupae and adults, although most expression was recorded in the larval alimentary canal. Importantly, tissue-specific Fo-G_N_ expression in the alimentary canal was verified by FISH assays. Third, dietary RNAi treatments led to steep declines in Fo-G_N_ expression over the following 24 h. Fourth, the RNAi treatments led to significant reductions in TSWV titers, documented by PCR and FISH assays. Fifth, the Fo-Cyp1 is quite similar to counterparts identified in several insect orders. Sixth, the Fo-Cyp1 identified here is very similar to cyclophilin genes in fruit fly and human genomes at the sequence and protein structure levels. Seventh, a detailed analysis of Fo-Cyp1 led to findings very similar to our analysis of Fo-G_N_, which we do not rehearse here. Taken together, these findings amount to a strong argument that both candidate genes act in TSWV entry and multiplication in *F. occidentalis*.

We note the Fo-G_N_ features a RR consensus, which occurs in a substantial proportion of arthropod cuticular proteins. Among its biological actions, chitin binding may be important [19]. Fo-G_N_ is an endocuticular protein (endoCP) which binds a TSWV coat protein [14]. However, the midgut does not have a cuticular structure because it originates from the endoderm during embryogenesis. Thysanopterans feature a lipoprotein membrane called the perimicrovillar membrane rather than chitin-containing peritrophic membrane [20]. We considered how the cuticular protein might interact with TSWV. Our study supports the idea that Fo-G_N_ is an endoCP due to its cuticle-binding domain, based on a bioinformatics analysis, and a gut protein, because its transcripts were recorded in our FISH assays. In addition to binding with a TSWV G_N_ coat protein, Fo-G_N_ was specifically expressed in young larvae along with differentially expressed genes after TSWV infection [21]. These points support our view of Fo-G_N_ as a TSWV receptor and silencing Fo-G_N_ expression led to reduced viral titers to near-zero, compared to control treatments.

Silencing Fo-Cyp1 expression prevented TSWV infection. CypA PPIase activity mediates the cis-trans interchange of adjacent side chains around cysteine residues, which is necessary in protein folding and protein-protein interactions [22]. Cyps share a common domain of approximately 109 amino acids called the cyclophilin-like domain (CLD), surrounded by domains unique to each member of the family [23,24]. They are found in all prokaryote and eukaryote cells. As a broad summary, humans have 16 Cyps, *Arabidopsis* have 29 Cyps, *Saccharomyces* have 8 Cyps, and *Drosophila* have 9 Cyps [25]. Cyclophilin A (CypA) is the first known Cyp in mammals and is the molecular target of the immunosuppressive drug, cyclosporine A [26]. Among the *F. occidentalis* Cyps, Fo-Cyp1 was proposed by another research group to interact with TSWV [14]. Phylogenetic analysis showed that Fo-Cyp1 is clustered with Cyp1 of *D. melanogaster*. Domain analysis indicates Fo-Cyp1 is located in the cytoplasm.

We infer successful TSWV infection is linked to PPI activity. We propose that TSWV may sequester Fo-Cyp1 to prevent an antiviral activity as seen in *Spodoptera litura*, infections of the *Microplitis bicoloratus* bracovirus (MbBV) [27]. MbBV infection induces translocation of CypA to the nucleus, where it may act in DNA fragmentation during apoptosis. *S. litura* CypA expression was upregulated after viral infection. Its suppression by RNAi significantly curtailed apoptosis of the target cells after MbBV infection. The CypA also acts in nuclear DNA fragmentation, noted by assessing its functional interaction with apoptosis-inducing factor (AIF) [28]. AIF is localized in the intermembrane region of mitochondria, and it translocates into the nucleus upon apoptotic signaling [29]. The CypA and AIF complex translocates into the nucleus in mammalian neurocytes, where the complex mediates DNA fragmentation [30]. MbBV infection increased AIF release from mitochondria and induced CypA expression. An in vitro binding assay showed the interaction of CypA and AIF. Blocking AIF release significantly inhibited translocation of the AIF/CypA complex to the nucleus and reversed apoptosis of the target cells infected by MbBV. These findings indicate CypA may have antiviral activity by mediating apoptosis in insect cells. In our interpretation, TSWV may interact with a host protein, Fo-Cyp1, for the dual purposes of enhancing infection and preventing Fo-Cyp1-induced antiviral responses. 

To transmit TSWV, young larvae must be infected with the virus by feeding on virus-infected plants [6,9]. After viral replication during larval development, TSWV particles accumulate in adult salivary glands, and they are transmitted to host plants via feeding [9]. TSWV acquisition from infected plants into larval midguts operates in TSWV dissemination.

These results indicate that Fo-G_N_ and Fo-Cyp1 are required for the circulative-propagative transmission of TSWV by the vector *F. occidentals*. This suggests to us that TSWV coat proteins interact with Fo-G_N_ on the gut of *F. occidentalis* larvae (Figure 7) because young larvae of *F. occidentalis* should be infected with the virus to transmit TSWV [6,31]. The viral G_N_ and Fo-G_N_ interaction may be associated with viral internalization in the epithelial cells. Within the cells, TSWV interacts with Fo-Cyp1 to suppress anti-viral responses to protect and multiply the viral particles. After the viral replication, TSWV particles accumulate in the principal salivary gland, especially in adults for viral transmission via feeding behavior [9]. Thus, the infection of TSWV to the larval midgut via Fo-G_N_ and Fo-Cyp1 is essential for TSWV dissemination.

Some virus-host plant-insect vector systems evolve rather convoluted relationships. Wang et al. [32] tested the idea that the Southern black-streaked dwarf virus (SRBSDV) influences the host plant preferences of its insect vectors, the brown plant hopper, *Nilaparvata lugens* (BPH) and the white-backed plant hopper (WBPH) *Sogatella furcifera*. The authors tested WBPH preferences for host plants that were either not infected or were infected with SRBSDV. They found that virus-free WBPH showed a marked preference for virus-infected host plants over virus-free plants. Similarly, the rice dwarf virus (RDV) influences the host plant preferences of its vector, the green rice leafhopper, *Nephotettix cincticeps* [33]. Again, virus-free leafhoppers preferred RDV-infected host plants, and virus-infected leafhoppers preferred RDV-free host plants. Both of these studies used olfactometers to test vector responses to host plant odors, finding the vector insects could discriminate between odors from virus-infected and non-infected host plants. These viruses influence vector preferences via altering host plant physiology. It is not yet known how many plant virus species influence their vector insects, nor whether TSWV also alters host plant preferences of its vector, *F. occidentalis*.

## Figures and Tables

**Figure 1 insects-14-00154-f001:**
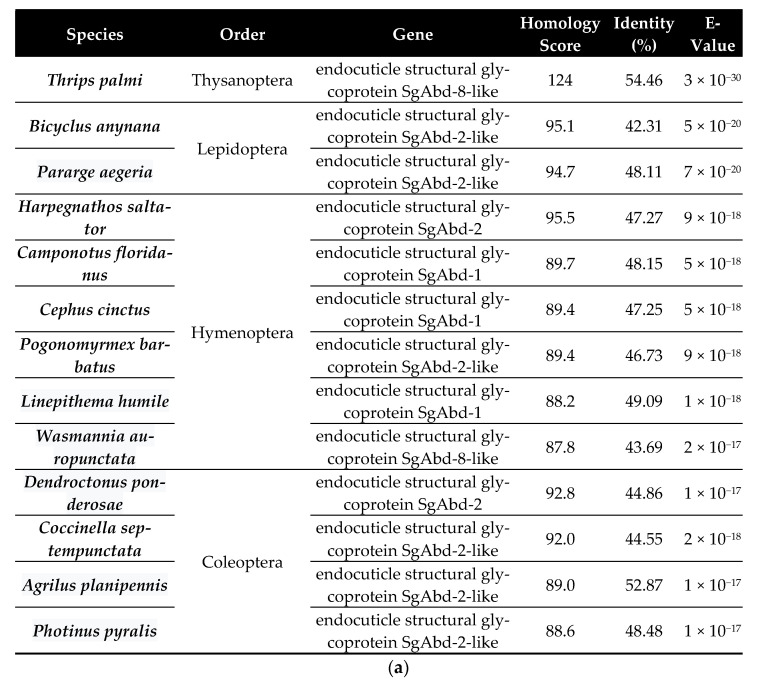
BLAST results and domain analysis of the *F. occidentalis* Fo-G_N_ gene. (**a**) Comparisons of the Fo-G_N_ gene between *F. occidentalis* and other insects from different orders. (**b**) Functional domain analysis of the *F. occidentalis* G_N_ gene. ‘SP’ denotes the signal peptide and ‘CHIT_BIND_4′ represents the Chitin Binding 4 domain with a RR consensus marked in red.

**Figure 2 insects-14-00154-f002:**
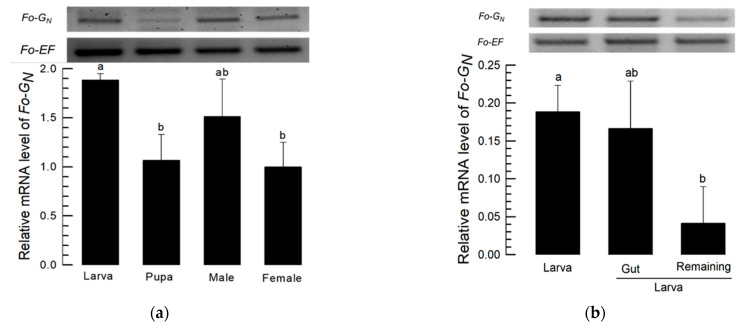
Fo-G_N_ expression profile. (**a**) Fo-G_N_ expression is higher in larvae and males and lower in pupae and females. (**b**) Within larvae, we recorded higher expression in whole larvae and isolated larval guts and significantly lower expression in gut-removed remaining larval tissues. Different letters above the standard deviation bars indicate significant differences among means at Type I error = 0.05 (LSD test). (**c**) Specific expression of Fo-G_N_ (green) in larval guts by FISH assay using anti-sense probe. In sense probe treatment, the contour is drawn because of no signal. FG, foregut; SG, salivary glands; MG, midgut; HG, hindgut and MT, Malpighian tubules. All scale bars are equal to 0.1 mm. G_N_, Glycoprotein.

**Figure 3 insects-14-00154-f003:**
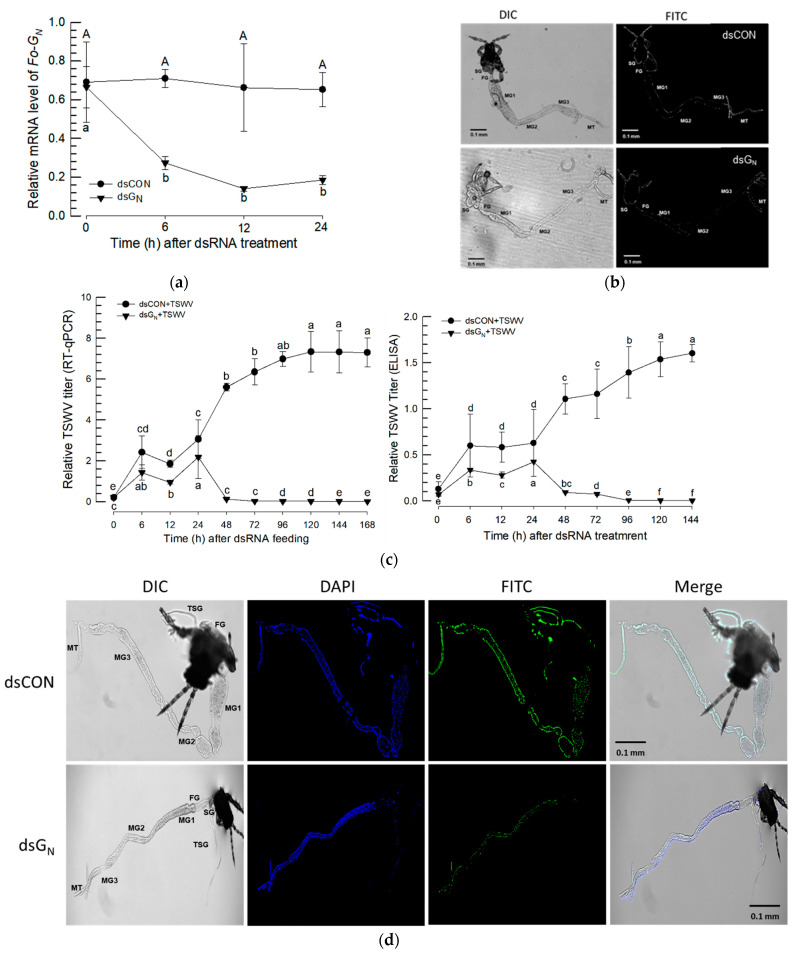
The influence of RNAi treatments on Fo-G_N_ expression. (**a**) Dietary RNAi led to decreased Fo-G_N_ expression in the L1 larva over the following 24 h. (**b**) FISH assays after dietary dsG_N_ show that Fo-G_N_ expression in the alimentary canal was not visible. For the FISH assay, the larval gut was dissected after dsRNA feeding and treated with FAM-labeled sense and anti-sense primers specific to Fo-G_N_. (**c**) Relative TSWV titer analysis using RT-qPCR and ELISA. The first instar larvae were fed on dsG_N_ and 12 h later, after dsRNA application, TSWV was applied. A non-target gene, *CpBV302*, was used as a control dsRNA (dsCON). Different letters above the standard deviation bars indicate significant difference among means in each treatment at Type I error = 0.05 (LSD test). Immunolocalization of tomato spotted wilt virus (TSWV) infection in the midgut and salivary gland of *F. occidentalis* in the larvae (**d**) and adults (**e**). FG, foregut; SG, salivary glands; MG, midgut; HG, hindgut and MT, Malpighian tubules. All scale bars are equal to 0.1 mm.

**Figure 4 insects-14-00154-f004:**
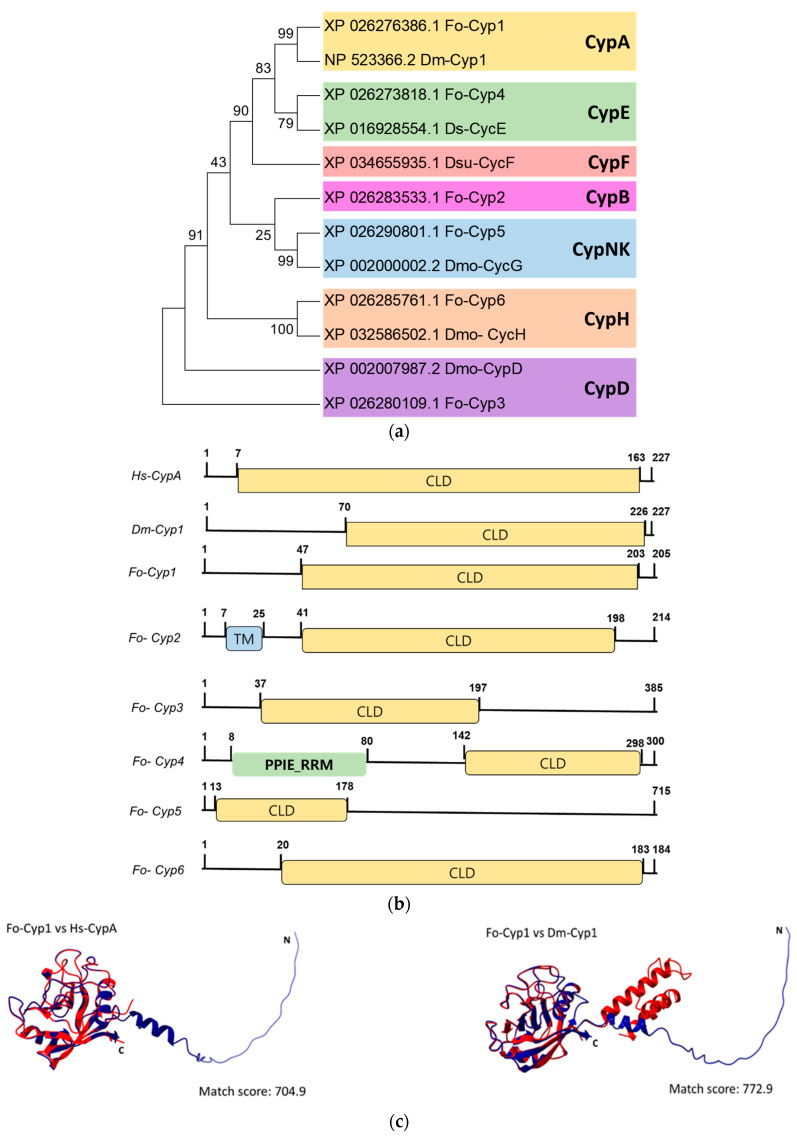
Molecular characterization of the *F. occidentalis* cyclophilin genes. (**a**) Phylogenetic analysis between selected cyclophilin genes present in fruit flies (*Drosophila* spp.) and WFTs (*F. occidentalis*) and compared with cyclophilin genes present in humans (*Homo sapiens*). The phylogenetic tree was generated by the Neighbor-joining method using MEGA6.0. Bootstrapping values were obtained with 1000 repetitions to support branch and clustering. Amino acid sequences were retrieved from GenBank. Gene accession numbers are shown in the figure. (**b**) Functional domain analysis and comparison of six cyclophilin genes present in *F. occidentalis*, the *Drosophila melanogaster* cyclophilin 1, and the *Homo sapiens* cyclophilin A. The predicted domain included “CLD” for the cyclophilin-like domain, “TM” for the transmembrane, and “PPIE_RRM” for the RNA recognition motif. Domains were predicted using EMBL-EBI (www.ebi.ac.uk accessed on 29 November 2022) and Pfam (http://pfam.xfam.org accessed on 29 November 2022). Transmembrane domains were predicted using DeepTMHMM (https://dtu.biolib.com/DeepTMHMM, accessed on 29 November 2022). (**c**) 3D structural analysis between *F. occidentalis* (blue) and *Homo sapiens* (red) cyclophilin, where the match score is 704.9 and *F. occidentalis* (blue) and *Drosophila melanogaster* (red) cyclophilin, where the match score is 772.9. Three-dimensional structure and superimposition analysis were performed by UCSF Chimera (http://www.cgl.ucsf.edu/chimera, accessed on 29 November 2022). (**d**) Possible cyclosporin binding site in the Fo-Cyp1 structure and amino acid sequence alignment of Fo-Cyp1, Dm-Cyp1, and Hs-CypA. The alignment was generated by the MegAlign multiple sequence alignment program using the ClustalW program. Conserved amino acid sequences show two α helices (blue bar) and eight β-barrel structure (red bar) for *F. occidentalis*, *D. melanogaster*, and *H. sapiens*.

**Figure 5 insects-14-00154-f005:**
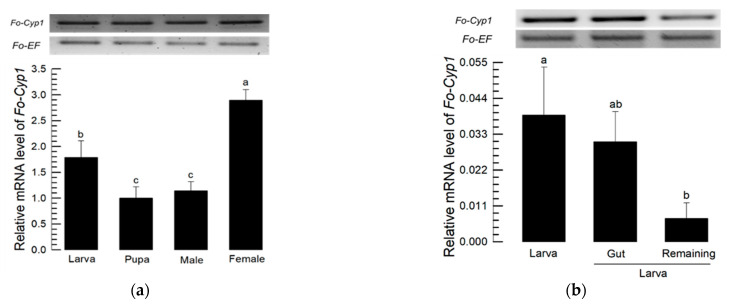
Expression profile of Fo-Cyp1. (**a**) Fo-Cyp1 expression in the indicated developmental stages. (**b**) Fo-Cyp1 expression in larvae, larval gut tissue, and remaining gut-removed tissues. An elongation factor, *EF1*, was used to normalize the expression level. Different letters above the standard deviation bars indicate significant differences among means at Type I error = 0.05 (LSD test). (**c**) FISH assays show Fo-Cyp1 expression (green) in the *F. occidentalis* larval gut using an anti-sense probe. In sense probe treatment, the contour is drawn because of no signal. The samples were prepared as described in the Section 2. FG, foregut; SG, salivary glands; MG, midgut; HG, hindgut and MT, Malpighian tubules. All scale bars represent 0.1 mm. Cyp1, cyclophilin 1.

**Figure 6 insects-14-00154-f006:**
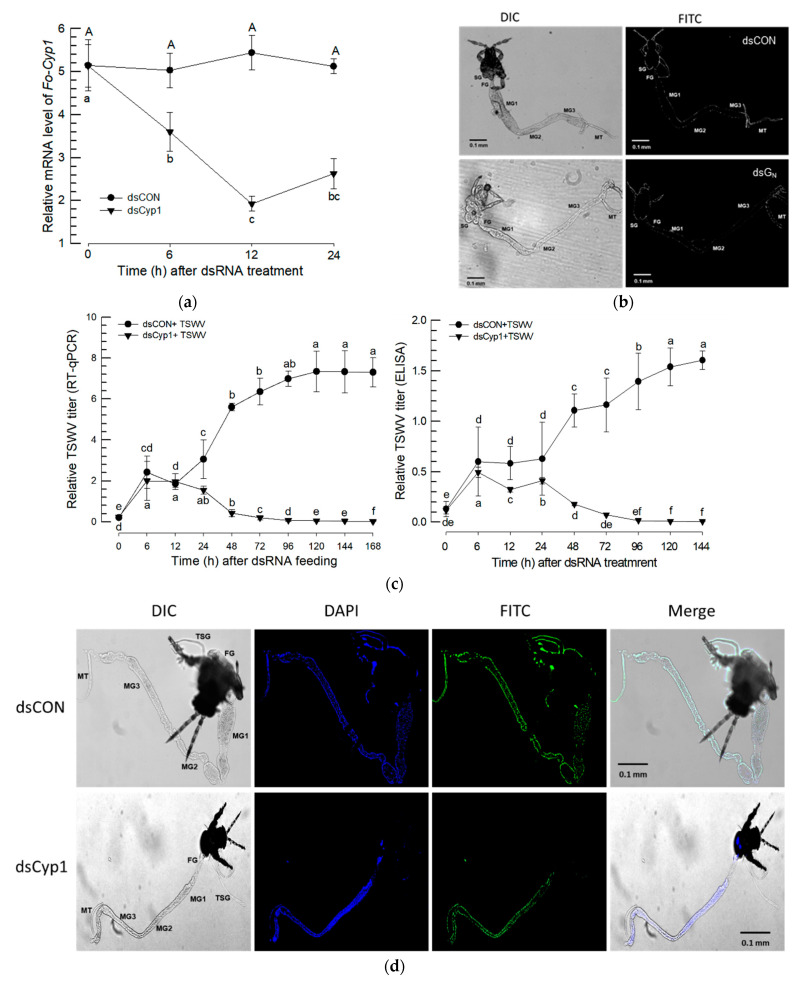
RNAi treatments using dietary dsRNA (dsCyp1) specific to Fo-Cyp1, detailed in the Section 2, led to a reduced Fo-Cyp1 expression in the gut and to a reduced TSWV titer. A non-target gene, *CpBV302*, was used as a control dsRNA (dsCON). (**a**) RNAi treatments led to reduced Fo-Cyp1 expression in L1 larvae. (**b**) FISH assays show the RNAi treatments abrogated the expression of Fo-Cyp1. (**c**) shows that the RNAi treatments led to reduced TSWV titres, determined by RT-qPCR and ELISA assays. Different letters above the standard deviation bars indicate the significant difference among means in each treatment at Type I error = 0.05 (LSD test). Immunolocalization of TSWV in the larval (**d**) and adult (**e**) midgut and salivary gland. FG, foregut; SG, salivary glands; MG, midgut; HG, hindgut and MT, Malpighian tubules. All scale bars are equal to 0.1 mm.

**Figure 7 insects-14-00154-f007:**
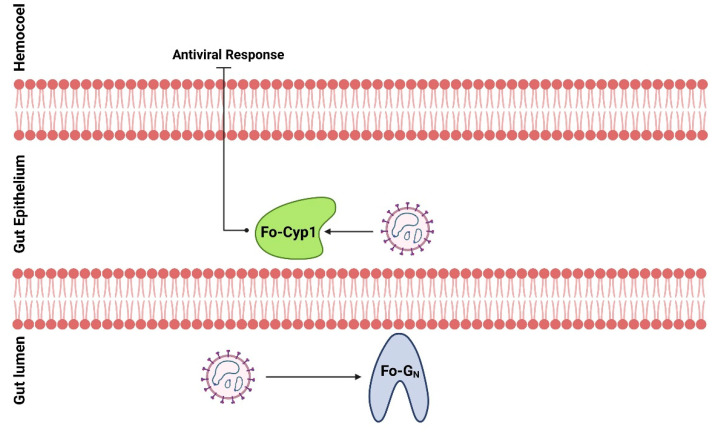
A working hypothesis of TSWV infection from the gut lumen to the gut epithelium of *F. occidentalis*. TSWV binds to a glycoprotein, Fo-G_N_, on the cell membrane to be internalized. The internalized TSWV binds to a cyclophilin, Fo-Cyp1, to prevent the induction of anti-viral responses.

## Data Availability

The data presented in this study are available in the article or Appendix A.

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
