# Peer review of "Two Alimentary Canal Proteins, Fo-GN and Fo-Cyp1, Act in Western Flower Thrips, Frankliniella occidentalis TSWV Infection"

_insects, 2023, doi:10.3390/insects14020154_

Round 1
Reviewer 1 Report
The authors have assessed the actions of two intestinal pro-teins, glycoprotein (Fo-GN) and cyclophilin (Fo-Cyp1) in their functional association with TSWV infection of the midgut, and found Silencing the two genes encoding these two genes led to near-zero reduc-tion of TSWV in midguts and salivary glands of the infected thrips. They proposed the two intestinal proteins, Fo-GN and Fo-Cyp1 are TSWV entry targets that are necessary to infect Frankliniella occidentalis for continued transmission to additional host plants.
The manuscript is well written. Overall, it is an interesting study that I believe will appeal to the broad readers of Insects.
There are several minor comments and suggestions:
Figures: In the lengends: should indicate which statistical analysis methods were used in the figure.
Figure 3c and Figure 6c: I suggest to label the result of statistical analysis in the figures
2.2: For phylogenetic analyses and phylogenetic tree construction, why Neighbor-joining method was used? In addition, was the optimal substitution model for the analysis evaluated before building the tree? Also, adding an additional phylogenetic analysis method should be better.
Figure 4a. Is it nessecery to add outgroups in this figure?
Author Response
Comment #1-1: Figures: In the lengends: should indicate which statistical analysis methods were used in the figure.
Response: In all statistical assays to compare the means, we used LSD test. This information was added to the figure captions as follows: “Different letters above standard deviation bars indicate significant differences among means at Type I error = 0.05 (LSD test).”
Comment #1-2: Figure 3c and Figure 6c: I suggest to label the result of statistical analysis in the figures
Response: The statistical analysis is performed to compare the mean differences. The results are added to the figures.
Comment #1-3: 2.2: For phylogenetic analyses and phylogenetic tree construction, why Neighbor-joining method was used? In addition, was the optimal substitution model for the analysis evaluated before building the tree? Also, adding an additional phylogenetic analysis method should be better.
Response: Our phylogeny tree was not to trace of the evolutionary trajectory. We used a phylogenetic analysis to determine the orthologs of the thrips cyclophilin.
Comment #1-4: Figure 4a. Is it nessecery to add outgroups in this figure?
Response: This phylogeny tree analysis was performed to determine the orthologs between the thrips cyclophilin and human cyclophilin (which has been well known). Thus we believe this tree does not need outgroup.
Reviewer 2 Report
Dear Authors,
I read your submitted mns Inects-2178298 and I was well impressed by the interesting study and its scientific soundness. Also the high quality of presentation. In particular , I consider quite good the Results and Discussion Sections (that critically consider , at the same time, the comments to the results obtained and their possible new developments). I think that the mns could be accepted in the present form , but there are a few minor suggestions I noticed in the attached word text ( i.e. in M&M section, at paragraphs 2.1 and 2.2. In the Results section at caption of Figure 4). See, please, these little notes.

Author Response
Comment #2-1: I read your submitted mns Inects-2178298 and I was well impressed by the interesting study and its scientific soundness. Also the high quality of presentation. In particular , I consider quite good the Results and Discussion Sections (that critically consider , at the same time, the comments to the results obtained and their possible new developments). I think that the mns could be accepted in the present form , but there are a few minor suggestions I noticed in the attached word text ( i.e. in M&M section, at paragraphs 2.1 and 2.2. In the Results section at caption of Figure 4). See, please, these little notes.
Response:
All reviewer’s suggestions are reflected in the revised version.
Reviewer 3 Report
The work by Falguni Khanand colleagues about Two alimentary canal proteins, Fo-GN and Fo-Cyp1, act in western flower thrips, Frankliniella occidentalis TSWV infection
I have found some minor details in the ms that need to be revised, hopefully improving the present work's quality.
1) Keywords: Sort key-words alphabetically.
2) In the description of the figures, abbreviations should not be used. 3) Introduction: The last paragraph must mention the objectives of your work. 4) Please more details for this section : 2.1. Insect rearing. And add section: SWV inoculum 5) Indicate the age of larvae and pupae used in different experiments.6) it is preferable to add graphical illustrating the internal anatomy of the salivary glands and digestive tract in relation to the intestine of Frankliniella occidentalis and the tissue tropism associated with larvae and adults during infection with tomato spotted wilt virus
Author Response
Comment #3-1: Keywords: Sort key-words alphabetically.
Response: Keywords are alphabetically re-ordered.
Comment #3-2: In the description of the figures, abbreviations should not be used.
Response: We tried to avoid abbreviations except LSD, ELISA, FAM. All abbreviations in the figures are explained in the captions.
Comment #3-3: Introduction: The last paragraph must mention the objectives of your work.
Response: The purpose is rephrased as follows: “In our approach to this issue, we tested the hypothesis that the candidate proteins Fo-GN and Fo-Cyp1 act in TSWV entry and multiplication in F. occidentalis.”
Comment #3-4: Please more details for this section : 2.1. Insect rearing. And add section: SWV inoculum
Response: For 2.1. Insect rearing, we changed the source of the test insects. For TSWV inoculation, it is written as follows: “For the F. occidentalis immune challenge, sprouted bean seed kernels were dipped into 1 mL of the viral suspension for 5 min and dried for 10 min under aseptic conditions. Treated bean seeds were placed in a circular breeding container (100 mm x 40 mm, SPL) and test insects were fed for 12 h, then the virus-treated bean seeds were replaced with the sprouted beans for the insects to be used in experiments.”
Comment #3-5: Indicate the age of larvae and pupae used in different experiments.
Response: As mentioned, TSWV inoculation was performed in first instar (L1) larvae. Adults were used in different time points to assess the virus propagation.
Comment #3-6: it is preferable to add graphical illustrating the internal anatomy of the salivary glands and digestive tract in relation to the intestine of Frankliniella occidentalis and the tissue tropism associated with larvae and adults during infection with tomato spotted wilt virus
Response: It is a nice suggestion. As you may know, the viral infection and the subsequent tissue tropism have been reported in previous studies. Thus, this study demonstrated the viral propagation before we assessed the target analysis. Instead, we add all detailed anatomical points are explained in the figure caption with abbreviations in the photos.